# Enzymatic and Molecular Characterization of Anti-*Leishmania* Molecules That Differently Target *Leishmania* and Mammalian eIF4A Proteins, LieIF4A and eIF4A_Mus_

**DOI:** 10.3390/molecules27185890

**Published:** 2022-09-10

**Authors:** Yosser Zina Abdelkrim, Emna Harigua-Souiai, Imen Bassoumi-Jamoussi, Mourad Barhoumi, Josette Banroques, Khadija Essafi-Benkhadir, Michael Nilges, Arnaud Blondel, N. Kyle Tanner, Ikram Guizani

**Affiliations:** 1Laboratory of Molecular Epidemiology and Experimental Pathology (LR11IPT04/LR16IPT04)/Laboratory of Epidemiology and Ecology of Parasites, Institut Pasteur de Tunis—University Tunis El Manar, Tunis 1002, Tunisia; 2Université de Paris Cité & CNRS, Expression Génétique Microbienne, Institut de Biologie Physico-Chimique, 13 Rue Pierre et Marie Curie, F-75005 Paris, France; 3Paris Sciences and Lettres Research University, Institut de Biologie Physico-Chimique, 13 Rue Pierre et Marie Curie, F-75005 Paris, France; 4Structural Bioinformatics Unit, Institut Pasteur, F-75015 Paris, France

**Keywords:** 7-α-aminocholesterol, drug design, translation-initiation factor, inhibitor, *Leishmania infantum*

## Abstract

Previous investigations of the *Leishmania infantum* eIF4A-like protein (LieIF4A) as a potential drug target delivered cholestanol derivatives inhibitors. Here, we investigated the mode of action of cholesterol derivatives as a novel scaffold structure of LieIF4A inhibitors on the RNA-dependent ATPase activity of LieIF4A and its mammalian ortholog (eIF4AI). We compared their biochemical effects on RNA-dependent ATPase activities of both proteins and investigated if rocaglamide, a known inhibitor of eIF4A, could affect LieIF4A as well. Kinetic measurements were conducted at different concentrations of ATP, of the compound and in the presence of saturating whole yeast RNA concentrations. Kinetic analyses showed different ATP binding affinities for the two enzymes as well as different sensitivities to 7-α-aminocholesterol and rocaglamide. The 7-α-aminocholesterol inhibited LieIF4A with a higher binding affinity relative to cholestanol analogs. Cholesterol, another tested sterol, had no effect on the ATPase activity of LieIF4A or eIF4AI. The 7-α-aminocholesterol demonstrated an anti-*Leishmania* activity on *L. infantum* promastigotes. Additionally, docking simulations explained the importance of the double bond between C5 and C6 in 7-*α*-aminocholesterol and the amino group in the C7 position. In conclusion, *Leishmania* and mammalian eIF4A proteins appeared to interact differently with effectors, thus making LieIF4A a potential drug against leishmaniases.

## 1. Introduction

Leishmaniases are caused by protozoan parasites from more than 20 *Leishmania* species, and they form a group of neglected tropical diseases. The epidemiology of Leishmaniases is complex, with many vectors and reservoir hosts incriminated in their transmission and spread. The parasites are transmitted to humans by the bite of an infected female phlebotomine sandfly. Leishmaniases consists of three main forms: cutaneous leishmaniasis (CL), visceral leishmaniasis (VL), also known as kala-azar, and mucocutaneous leishmaniasis (MCL). CL is the most common form, VL is the most severe form, and MCL is the most disabling form of the disease. In 2018, the number of countries or territories considered endemic for or with previously reported cases of CL and VL were 92 and 83, respectively. Today, more than 1 billion people live in areas endemic to leishmaniasis and are at risk of infection. An estimated 30,000 new cases of VL and more than 1 million new cases of CL annually occur. Hence, leishmaniases are considered a serious public health problem [1]. Several vaccine research programs to fight leishmaniases are being pursued [2]. However, there is currently no vaccine against any form of the disease approved in humans despite efforts using parasite or vector targets. Correspondingly, treatment options are unfortunately limited due to the lack of efficient drugs, the high costs and significant adverse effects. Another threat to leishmaniasis treatment is the apparition of drug resistance, as observed in multiple endemic countries [3,4].

The identification of novel drugs and therapeutic molecules constitutes a global research priority, particularly against the fatal VL form mainly caused by *L. donovani* and *L. infantum*. Different potential targets have been investigated [5,6,7]. Several criteria can characterize potential drug targets. They notably include genetic attributes, such as expression at relevant life stages and essentiality, as well as biochemical properties and features influencing druggability, assayability and the ability to identify inhibitors. For that purpose, we previously used *L. infantum* eIF4A (LieIF4A), a probable translation initiation factor belonging to the DEAD-box proteins family of RNA helicases, as a potential drug target [7,8]. This protein, as we previously noted, is encoded in the annotated genomes by two *Leishmania* eIF4A tandem versions (LINF_010012800 and LINF_010012900 in *L. infantum*) encoding identical proteins [9,10]. DEAD-box RNA helicases are ubiquitous proteins found in all kingdoms of life and are associated with all processes involving RNA, from transcription, processing, transport and translation to RNA decay [11,12]. They have been shown to disrupt short RNA-RNA and RNA-DNA duplexes, displace RNA-bound proteins, remodel ribonucleoprotein complexes, act as RNA chaperones to facilitate the formation of functional RNA secondary and tertiary structures, and also act as ATP-dependent RNA clamps to facilitate multistep biological reactions [12,13]. The eIF4A proteins are particularly thought to unwind secondary structures impeding scanning ribosomes, but recent cryo-electron microscopy has shown that the eIF4F complex is actually behind the scanning ribosomes, and thus it may function as an ATP-dependent clamp to prevent backward sliding of the scanning ribosomes [14]. In vitro, they are ATP-dependent RNA-binding proteins and RNA-dependent ATPases, but they generally lack RNA substrate specificity. In vivo, DEAD-box proteins are highly specific and regulated. The specificity is probably conferred by the cellular context and associated cofactors [15]. The DEAD-box proteins have a core structure consisting of two domains having structural homology with recombinant protein A (RecA) connected by a linker region [16]. They share conserved motifs associated with ligand binding and enzymatic activity, which are located within this core domain, notably, the Walker B motif (motif II) with a typical amino-acid sequence of D-E-A-D [11,12,17]. Motifs Ia, GG, Ib, IV, QxxR and V are involved in RNA binding. The Q motif recognizes the adenine, motifs I (Walker A motif), II (Walker B motif), III, V and VI interact with the phosphate residues of ATP. The binding of ATP and RNA substrates promote a “closed” conformation of the two RecA-like domains, which have a high affinity for the ligands. An “open” conformation with low affinity, in which the RecA-like domains are not constrained relative to one another, is adopted once the ATP is hydrolyzed and ADP and inorganic phosphate are released [12,13,17,18,19]. RNA helicases represent important targets for therapeutic and prophylactic drugs because of their central role in RNA metabolism and function [8,20]. Particularly, the mammalian DEAD-box proteins eIF4A have been of particular interest for screening drug candidates [21]. These proteins show sequence and structural differences despite their shared features [22,23].

To respond to the urgent need for novel treatments, different drug targeting strategies and drug sources are used. Naturally occurring plants offer a source for screening and isolating bioactive compounds endowed with antileishmanial activities as a lead to develop potential therapeutic molecules [24,25,26,27,28]. Exploring natural products from microorganism origins also remains a valuable approach to finding novel hits and candidates for the development of new drugs against *Leishmania* human infections [29,30,31]. Recently, a natural and bioactive triterpene celastrol was used as a lead compound for the design and synthesis of a bioinspired, novel potential leishmanicidal agent. The authors designed new compounds based on the modulation of the ATPase activity of *L. braziliensis* Hsp90 and on *Leishmania* growth inhibition [32].

Rocaglamide (also called rocaglamide A or Roc-A) is a natural insecticide isolated from the dried roots and stems of *Aglaia elliptifolia Merr*. The plant was identified as an anti-leishmanial compound that inhibited *L. infantum* promastigotes with an EC50 of 16.5 µM, although not without hepatotoxic activity in human primary hepatocytes [33]. However, this seems to be unspecific because antiviral activity against several viruses for other rocaglates was calculated at low nanomolar concentrations [34]. It had shown antiproliferative activity against various human cancer cell lines at nanomolar concentrations [35]. It was shown to have an antileukemic activity in mice and an inhibitory activity in cells derived from human epidermoid carcinoma [23,35,36]. Several naturally occurring rocaglamide derivatives belonging to the cyclopenta[b]benzofuran class of molecules [35] were shown to inhibit translation [37]. Sadlish et al. found a discrete set of residues within the RNA binding motif Ib and between motifs II and III of RecA-like domain 1 of the yeast eIF4Aprotein that conferred resistance to synthetic rocaglamide-derivative ROC-N when mutated [38]. They demonstrated that the tested compounds enhanced RNA binding and stimulated the RNA-dependent ATPase and ATP-dependent unwinding activities in wildtype eIF4A but not in the mutants [38].

The limited number of available therapeutic options emphasizes the need to discover novel leishmaniases chemical treatments. One way to accelerate drug discovery is to use virtual screening exploiting available structural information to allow coverage of a large chemical space and narrowing down the number of candidates for the subsequent experimental validations.

Cholestanol derivatives were previously identified as anti-*Leishmania* molecules through virtual, biochemical and biological screenings [7]. Our group used molecular modeling and molecular dynamic simulations of LieIF4A [7,9,39,40] to identify potential pockets on the protein, and then we screened potential candidates for selected sites. We discovered novel cholestanol derivative inhibitors (6-α-aminocholestanol and 6-ketocholestanol) displaying LieIF4A ATPase inhibition and anti-leishmanial effects, both on promastigote and amastigote *L. infantum* forms; these results consequently validated LieIF4A as a potential drug target for chemicals [7]. The 6-α-aminocholestanol and 6-ketocholestanol compounds exhibited anti-leishmanial activities with IC_50_ values of 3.6 µM and 39.1 µM, respectively, that were positively correlated with those determined for LieIF4A ATPase assays (160 μM and > 1 mM, respectively). They also inhibited the ATPase activity of LieIF4A and mouse eIF4AI (eIF4AI_Mus_) with different kinetic properties [7]. The 6-ketocholestanol presented a ketone group instead of the amino group on carbon C6 of 6-α-aminocholestanol. Eight tested analogs lacking this group were non-active and replacing it with a nitro group in the same position did not restore inhibition. Thus, the amino group appeared to be key for the inhibitory activity [7]. We have further characterized the interaction of these drugs with both human and *Leishmania* eIF4A at the molecular level and found evidence that the two proteins interacted differently with these compounds [8]. We also found that 6-α-aminocholestanol had a dual effect; it inhibited both the ATPase and helicase activities through a perturbation of ATP and RNA binding. Kinetic analyses showed that the *Leishmania* LieIF4A protein bound 6-α-aminocholestanol with a higher apparent affinity than for ATP, although multiple binding sites were probably involved. Competition experiments with the individual RecA-like domains indicated that the primary binding sites were on RecA-like domain 1, and they included a cavity previously identified by molecular modeling of LieIF4A that involved conserved RNA-binding motifs [8].

Bazin et al. investigated the sterol pathway by considering the similarities in the sterol biosynthesis pathways between fungi and *Leishmania* to synthesize oxysterols and nitrogenous sterol inhibitors [41]. These compounds showed anti-leishmanial activity against *L. donovani* parasites. The most active compounds against *L. donovani* promastigotes were 7-β-aminomethylcholesterol and 7-α/β-aminocholesterol (IC_50_ in a range from 1 to 3 μM). These compounds were active on intramacrophage amastigotes with an IC_50_ value of 1.3 μM. The authors suggested that they mediated their antileishmanial activity by interfering with sterol metabolism. However, no molecular or experimental data were provided to identify the potential target in the parasite [41]. The compound 7-aminocholesterol was initially described for its ability to inhibit yeast cell growth without complete inhibition of ergosterol production, indicating a different mechanism than impairment of sterol synthesis [42]. The 7-α/β-aminocholesterol also displayed an antibiotic activity against Gram-positive bacteria [43]. The wide activity spectrum of 7-α/β−aminocholesterol, notably the anti-leishmanial activity against *L. donovani*, and the structural similarities of the 7-α-aminocholesterol and 6-α-aminocholestanol (Figure 1), previously described as an inhibitor of LieIF4A, prompted us to investigate whether eIF4A proteins could be targets for cholesterol derivatives and if they would also differentially interact with these compounds and other known eIF4A inhibitors such as rocaglamide.

We performed enzymatic and in silico comparisons of the effects of cholesterol and cholestanol derivatives on LieIF4A and a mammalian eIF4A. Actually, we used the eIF4A_Mus_ gene, which encodes the identical amino acid sequence as human eIF4AI (DDX2A). We demonstrated different biochemical effects of cholesterol derivatives (7-*α*-aminocholesterol and cholesterol) on *Leishmania* and mammalian eIF4A ATPase activities and showed the higher inhibitory efficiency and specificity of the 7-*α*-aminocholesterol towards *Leishmania* eIF4A relative to other cholesterol and cholestanol derivatives tested in the previous and current studies [7,8]. Furthermore, through enzymatic and molecular characterization, we revealed important structural features of amino steroid derivatives, 7-*α*-aminocholesterol and 6-*α*-aminocholestanol for the inhibition of LieIF4A. Our study revealed a strong correlation between the presence of the amino group and the double bond between carbons 5 and 6 in 7-*α*-aminocholesterol that made the ring structure more planar and that both conferred the inhibitory efficiency and specificity towards *Leishmania* eIF4A in vitro. Briefly, we showed that the lack of the amino group on the 7-*α*-aminocholesterol abolished the inhibitory effect on the ATPase activity of LieIF4A and eIF4A_Mus_ and reduced the interactions of the compound with the *Leishmania* protein. The higher inhibition efficacy of the amino-derived compounds, when compared with their analogs, was consistent with the presence of an H-bond commonly established between the Asp332 residue from motif V of the *Leishmania* protein and the amino groups of 7-α-aminocholesterol and 6-α-aminocholestanol. Thus, all these structural features appeared crucial for the inhibitory effect on the promastigote viability of *L. infantum*. We also assessed the biochemical affinities of the *Leishmania* and mammalian eIF4A proteins towards the ATP substrate. In addition, we further compared the effect of rocaglamide, a known translation inhibitor targeting eIF4A that was also previously identified as an anti-leishmanial compound that inhibits *L. infantum* promastigotes [33] on the ATPase activity of the two proteins.

Importantly, we demonstrated markedly different responses from *Leishmania* and mammalian eIF4A proteins, which interact with substrates and compounds with different affinity and sensitivities. Overall, our results indicated that *Leishmania* and mammalian eIF4A proteins interacted differently with effectors (stimulators or inhibitors) and could thus be considered as different operational proteins. Notably, this work provided further evidence that chemicals could distinctively target *Leishmania* and mammalian eIF4A proteins and consequently endorsed the potential of LieIF4A as a drug target against Leishmaniases.

## 2. Results

### 2.1. eIF4A_Mus_ and LieIF4A Have Different Enzymatic Parameters for ATP Hydrolysis

We previously characterized the ATPase activity of LieIF4A protein and determined its Michaelis-Menten parameters in the presence of saturating concentrations of total yeast RNA, control DMSO and in the absence of a test compound. We obtained a K_m_ of 150 ± 20 μM (standard error of the regression) [8]. Experiments were also done for eIF4A_Mus_ at the same time by the same investigators in the same laboratory. However, kinetic parameters were not determined for the eIF4A_Mus_. Here, we determined them for comparison purposes under the same conditions. We found that the rate of phosphate release during ATP hydrolysis by eIF4A_Mus_ in the presence of saturating concentrations of total yeast RNA and control DMSO increased with the concentration of ATP (Figure 2), and we obtained a K_m_ value of 370 ± 49 µM (standard error of the regression), a lower affinity than for LieIF4A [8]. Moreover, the k_cat_ value (V_max_/[eIF4A_Mus_]) was 1.54 ± 0.07 min^−1^, which was 56% slower than for LieIF4A under the same conditions (2.4 ± 0.1 min^−1^) [8]. However, these values were in the same range as those obtained by other authors for mice or other organisms [9,44]. This demonstrated that under similar conditions, eIF4A_Mus_ and LieIF4A have different enzymatic parameters for ATP hydrolysis, notably a lower ATP binding affinity and a slower catalytic activity of eIF4A_Mus_ than LieIF4A.

### 2.2. Rocaglamide A Affects LieIF4A and eIF4A_Mus_ Differently

Rocaglamide A, a known eIF4A inhibitor affecting RNA binding and also described as an anti-Leishmania molecule [33], was shown to stabilize yeast eIF4A-RNA interactions and increase the ATPase activity by binding to a pocket [7,38] close to that we predicted for 6-α-aminocholestanol that exhibits binding to the RNA-binding motifs of LieIF4A [8]. Therefore, we investigated the effect of rocaglamide A on the ATPase activity of LieIF4A. However, LieIF4A was 83% (1.76/0.96) more active than eIF4A_Mus_ in the presence of 1 mM ATP and saturating concentrations of RNA (Table 1). Therefore, to facilitate comparisons and maximize the sensitivity of the assay, we titrated the RNA concentrations in the ATPase assays to obtain similar activity under non-saturating RNA conditions. We observed that eIF4A_Mus_ in the presence of 55 ng/µL of yeast RNA had comparable activity to that of LieIF4A with 1 ng/µL of RNA (Table 1). This suggested either that eIF4A_Mus_ had a much weaker RNA binding affinity than LieIF4A or that eIF4A_Mus_ bound the RNA in a less productive (less active) state. The latter possibility appears reasonable since the RNA-dependent ATPase activity of eIF4A_Mus_ might be reduced by its conformational state, as reported for the DEAD-box protein, DDX19, where the amino-terminal extension negatively regulates the activity [45].

If rocaglamide A similarly increased the affinity for RNA for eIF4A_Mus_ and LieIF4A, as observed for yeast eIF4A, it should also increase their ATPase activities to similar extents. However, 50 µM of rocaglamide A increased the ATPase activity of eIF4A_Mus_ by 6.2-fold (2.55/0.41), nearly 2.7-fold more (2.55/0.96) than with saturating concentrations of RNA, while the increase was only 1.7-fold for LieIF4A (0.77/0.47), 2.3-fold less (1.76/0.77) than with saturating concentrations of RNA (Table 1). Noticeably, rocaglamide A activated eIF4A_Mus_ to the same molar activity (2.55/1.76 × 820/1200 ≈ 1) as the maximal activity of LieIF4A without the compound but at saturating RNA concentrations. Hence, rocaglamide A induced an increase in intrinsic ATPase activity and probably RNA binding of eIF4A_Mus_ as compared with LieIF4A, possibly by some distinct allosteric effect. Hence, eIF4A_Mus_ and LieIF4A, which are from different organisms but implicated in similar processes, could have very different sensitivities to a drug like rocaglamide A.

### 2.3. Cholesterol Derivatives Affected the ATPase Activity of LieIF4A and eIF4A_Mus_ Differently

Virtual and biochemical screenings selected cholestanol derivatives as novel inhibitors of eIFA4_Mus_ and LieIF4A ATPases; compounds lacking the amino group in their structure were less (or not) effective than 6-α-aminocholestanol, which suggested the importance of the amino group for the inhibitory activity of the cholestanol derivatives. Cholesterol derivatives, which are structurally close to cholestanol, have proven anti-leishmanial activity, supposedly affecting the sterol biosynthesis [41]. Therefore, in this study, we asked whether sterol derivatives, namely 7-α-aminocholesterol and cholesterol, affected Leishmania and mammal eIF4A proteins. We evaluated the contribution of the amino group of 7-α-aminocholesterol to the affinity for the proteins. Notably, 7-α-aminocholesterol contained a 5–6 position double bond and an amino group at a different position than in 6-α-aminocholestanol.

We measured ATPase kinetics for LieIF4A and eIF4A_Mus_ at compound concentrations ranging from 0 to 300 µM. The velocity of hydrolyzed ATP for both proteins allowed us to determine the ATPase reaction rates at each compound concentration. Increasing concentrations of the compound decreased the ATPase activity (Figure 3). In contrast, the ATPase kinetics for both proteins in the presence of 300 µM cholesterol, which lacked the amino group, showed no effect on either LieIF4A or eIF4A_Mus_. This established LieIF4A and eIF4A_Mus_ as targets of the cholesterol derivatives and revealed the crucial role of the amino group.

The K_i_ values for cholesterol and cholestanol derivatives, calculated from kinetic analyses, are shown in Table 2. K_i_, similar to K_m_, represents the binding affinity of the inhibitor to the protein, and these constants can be compared. Interestingly, the 7-α-aminocholesterol compound showed its highest ATPase inhibition activity for LieIF4A with a K_i_ value of 8.6 µM, and a ~9.4-fold higher affinity than for eIF4A_Mus_ (80.5 μM). In comparison, the K_i_ values reported for the cholestanol derivatives were similar for both proteins. For 6-α-aminocholestanol, the best inhibitor after 7-α-aminocholesterol, the K_i_ value was 19.5 μM for LieIF4A and 20.2 μM for eIF4A_Mus_. Thus, 7-α-aminocholesterol specifically inhibited LieIF4A with a significantly higher affinity than for eIF4A_Mus_ and with a stronger inhibitory activity on LieIF4A than 6-α-aminocholestanol.

### 2.4. The Compound 7-α-Aminocholesterol Showed a Better Anti-Leishmania Activity against L. Infantum Promastigotes Than 6-α-Aminocholestanol Consistent with LieIF4A ATPase Inhibition

As a proof of concept, we evaluated the effect of the tested sterols on the viability of stationary phase L. infantum promastigotes using an MTT assay after a 24 h exposure. The 7-α-aminocholesterol, which inhibited the ATPase activity of LieIF4A, affected the promastigote viability in a dose-dependent manner. The 6-ketocholestanol was used as a control. We calculated the IC_50_ values (Table 3) and obtained 2.29 μM and 37 μM for 7-α-aminocholesterol and 6-ketocholestanol compounds, respectively. Cholesterol, which lacks the amino group, had no effect on promastigotes viability nor on the ATPase activities. Thus, enzymatic and biological results showed consistent trends.

The tested compounds revealed no significant toxicity on the THP-1 macrophages. The viability of the macrophages treated with each compound was ~90–100% at the concentration corresponding to the IC_50_ on the promastigotes (Figure 4). The CC_50_ values were calculated with GraphPad software as 17.5 μM and 77 μM for 7-α-aminocholesterol and 6-ketocholestanol compounds, respectively. Cholesterol did not affect the viability of the macrophages treated even at 100 µM.

### 2.5. Docking Experiments Revealed Plausible Active Moieties and Conformation Specificity for the Sterol Derivative 7-α-Aminocholesterol Binding the Leishmania Protein

The 3D structure of the LieIF4A protein, obtained through comparative modeling, served as a basis for a successful virtual screening of 305 molecules [7]. This 3D-structure model was further explored to identify an ensemble of pockets that may serve as potential binding sites for a series of chemical analogs of 6-α-aminocholestanol and was supported by in-depth biochemical analysis [8]. Four out of five pockets appeared to cover different subsites of the interdomain cleft of LieIF4A. In this respect, we ran our docking simulations targeting a space within the LieIF4A structure that included most of the key amino acids of the previously defined pockets, which also included residues defining the conserved motifs of the RNA helicase proteins. Molecular dockings of the 7-α-aminocholesterol, 6-α-aminocholestanol, 6-ketocholestanol and cholesterol were performed.

First, we retrieved the list of residues consistently positioned within 3.0 Å of the docked compounds on LieIF4A that included F28, F46, K48, P49, S50 from the Q motif, T74, T77 from motif I, E235, L237 from the RecA domain linker, R329, G330, D332 from motif V, and R357 from motif VI. These amino acids are involved in binding the ATP and RNA, and they are implicated in the enzymatic activity [12,17], supporting the docking pocket selection to target the RNA-dependent ATPase activity.

As shown in Figure 5, 6-α-aminocholestanol and 6-ketocholestanol adopted similar poses, while cholesterol adopted a similar docking pose to 7-α-aminocholesterol. The difference in the poses adopted by cholesterol versus cholestanol derivatives was correlated with the presence of the double bond between carbons 5 and 6 in the sterols (Figure 1), inducing a more planar form of their ring structure as compared to the cholestanols.

The scores of the selected docking poses showed that 7-α-aminocholesterol and 6-α-aminocholestanol had the best estimated binding energies with LieIF4A, which confirmed the experimental results (Table 4). The number of poses within the retained clusters brought additional evidence for the plausibility of the selected poses. Noticeably, the cycle containing carbons 6 and 7 occupied the same spatial location on all docking poses (Figure 5). The amino groups on carbons 7 and 6 of 7-α-aminocholesterol and 6-α-aminocholestanol compounds, respectively, established H-bonds with Asp332 (D332) from motif V (Figure 5). An interaction diagram to highlight the protein-inhibitor interactions is indicated in Figure 6, displaying the established contacts (H-bonds and hydrophobic) between the docked molecules and the protein. These interactions were not observed for 6-ketocholestanol, as it presented a keto group at position 6 (Figure 1). Cholesterol had no functional group at this position and did not present any equivalent H-bond at this position. These findings rationalized the correlation between the presence of the amino group and the inhibitory effect on the LieIF4A enzymatic activity.

## 3. Discussion

Leishmaniases are vector-borne parasitic diseases caused by *Leishmania* species, which inflict cutaneous, mucocutaneous or visceral diseases depending on the *Leishmania* species and the host immune response. Unfortunately, no vaccine is currently available, and the treatment relies mostly on antimonials and a few other drugs. The side effects and toxicity of most of the currently used drugs, as well as the emergence of drug resistance and therapeutic failures, create an important need to develop new antileishmanial drugs [3,4]. The discovery and development of antileishmanial drugs require, as for other drug discovery endeavours, the identification of potential drug candidates. These molecules can originate from natural sources or from chemical libraries, either from pharmaceutical companies, public access resources or small collections from academia [46]. Through in silico approaches, bioinformatics and experimental validations, we previously presented chemical evidence validating the *L. infantum* translation initiation factor eIF4A (LieIF4A) as a potential drug target against Leishmaniases. In particular, we identified promising cholestanol derivatives inhibitors. These compounds also inhibited *L. infantum* promastigotes and amastigotes growth and viabilities [7]. The most promising one, 6-α-aminocholestanol, inhibited the ATP and the RNA binding to LieIF4A and also the helicase activity; the primary binding sites are on RecA-like domain 1 [8]. The 6-α-aminocholestanol shows scaffold structure similarity to 7-α-aminocholesterol previously described for its wide spectrum properties as for its antileishmanial activity on promastigotes and intramacrophage amastigotes of *L. donovani* (LV9) but without data provided on its possible targets [41,42].

Cholesterol derivatives, namely 7-α-aminocholesterol and cholesterol, were selected in this study for their structural similarity with cholestanol derivatives, suggesting they could be eIF4A effectors, and they could thus interfere with LieIF4A activity and subsequently with *L. infantum* parasites viability. Hence, this study aimed at elucidating the effect of these compounds, as well as of known effectors such as rocaglamide, on eIF4A-like proteins, thereby validating them further as anti-leishmaniases targets. For that, we investigated whether LieIF4A and mammalian eIF4AI could be differentially affected to support the perspective of specific, non-toxic drug identification. Importantly, our results provided further biochemical evidence that the *Leishmania* and mammalian orthologues can have noticeably different sensitivities to chemical effectors or drugs, which is also supported and rationalized by molecular dockings of the compounds on LieIF4A.

We found that eIF4A_Mus_ binds ATP as a substrate with a similar K_m_ value as previously obtained for other organisms [9,44]. This binding was weaker than previously described for LieIF4A, and the catalytic activity was lower [8]. Thus, ATP binding affinities were one of the differences between the two enzymes. Further biochemical comparisons indicated that rocaglamide, which was previously described as an anti-*Leishmania* molecule [33], differently affected *Leishmania* and mammalian eIF4A proteins by stimulating their RNA-dependent ATPase activities to different extents, suggesting some differences in a possible allosteric mechanism. Previous findings proposed that rocaglamide inhibited translation pathways by stabilizing eIF4A-RNA interactions in yeast, consequently increasing the ATPase activity by binding to a pocket close to that predicted for 6-α-aminocholestanol [7,38]. The limited effects of rocaglamide on LieF4A, despite its antileishmanial effects, could point to the inhibition of other metabolic processes besides the eIF4A-like proteins in translation, at least for *Leishmania*.

Moreover, 7-α-aminocholesterol clearly had very different enzymatic effects on the two proteins. This compound presented higher specificity to the *Leishmania* protein compared with our previous results with 6-α-aminocholestanol and 6-ketocholestanol [7]. Cholesterol did not affect the enzymatic activity of either protein, revealing the importance of the amino group in the inhibitory activity of the compounds. Consistent with biochemical results, our biological investigations showed stronger effects of 7-α-aminocholesterol and 6-α-aminocholestanol on the *L. infantum* promastigotes viability (2.29 and 3.54 µM, respectively) than the other tested derivatives. In that respect, the specificity of 7-α-aminocholesterol was particularly interesting as it supports the perspective of differentially designed effectors.

In a previous study, we explained the differences in IC_50_ values observed for the ATPase activity and the promastigotes by different possible factors, including the high concentration in the in vitro ATPase reaction and the fact that *in cellulo* LieIF4A could be engaged in multimeric complexes [7]. Moreover, different studies have shown that the enzymatic activity of yeast and mammalian eIF4A was enhanced by translation factors, like eIF4F, eIF4B, eIF4H and eIF4G [47,48,49] without otherwise altering the enzyme properties. We previously showed that the ATPase activity of recombinant LieIF4A was slightly enhanced by yeast eIF4G [9]. Thus, the complex formation might change its activity without changing the basic binding characteristics of the inhibitors. Here, we tested the compounds on the individual proteins to simplify the enzymatic characterization, even though it would be interesting to further evaluate a reconstituted eIF4F complex in vitro. Moreover, the amounts of protein used in the assay were far above physiological concentrations determined in *Leishmania* [40]. All these observations reassured us that the inhibitory effect of the studied compounds would be efficient in the physiological conditions as well. Moreover, the in vitro antileishmanial activity of 7-α-aminocholesterol against intramacrophage amastigotes of *L. donovani* (LV9) was studied, and it exhibited an IC_50_ value of 1.31 µM [41]. Thus, in the future, it may be useful to additionally assess the effect of the cholesterol derivatives on *L. infantum* (LV50) amastigotes viability. All these observations also prompted us to screen a wide variety of amino steroid derivatives to identify more efficient and selective inhibitors of the *Leishmania* eIF4A protein.

Interestingly, molecular docking results were in line with the experimental data, and they provided a rationale for the inhibitory effects. The 6-α-aminocholestanol and 7-α-aminocholesterol docked in certain sites with a clearly discriminating score. Our docking simulations focused on a zone of LieIF4A that included key residues of the conserved motifs of the RNA helicase [8]. They revealed H-bonds between Asp332 from motif V involved in RNA binding [12,19] to the amino groups on carbons 7 and 6 of 7-α-aminocholesterol and 6-α-aminocholestanol compounds, respectively. This was not observed for the less active ATPase inhibitor 6-ketocholestanol, and the inactive cholesterol both lacked the amino group. Thus, the docking results correlated well with the biochemical, enzymatic and biological findings and suggested a specific role for the amino group and the global conformation of the molecule impacted by the location of the double bond between carbons 5 and 6 in the case of 7-*α*- aminocholesterol.

## 4. Materials and Methods

### 4.1. Compounds

Sterols, 7-α-aminocholesterol, cholesterol, 6-ketocholestanol and rocaglamide A were purchased from Sigma-Aldrich (St. Louis, MO, USA) under the references R207578, C3045, K1250 and SML0656, respectively. The structures of the compounds used are shown in Figure 1. The compounds were dissolved in dimethyl sulfoxide (DMSO; Sigma-Aldrich, St. Louis, MO, USA) to make stock solutions.

### 4.2. Protein Expression and Purification

The purified His6-tagged LieIF4A and eIF4A_Mus_ protein stocks were previously prepared and used in our published works [7,8]. In brief, the proteins were prepared as previously described from already constructed recombinant expression vectors [8,9,50]. They were expressed in the Rosetta *Escherichia coli* strain upon induction with 0.5 mM IPTG, and the proteins were purified using nickel-affinity chromatography. The pellets collected by cell centrifugation were lysed using a lysis buffer (Tris-HCl, pH 7.5, 300 mM NaCl, 10 mM imidazole and 0.02% NP40) supplemented with a lysozyme and protease inhibitor cocktail. After the sonication and centrifugation of the cells (30 min, 15,000 rpm in a JA-20 rotor (Beckman Coulter, Villepinte, France), 4 °C), the supernatant was loaded onto a 1 mL nickel-nitrilotriacetic acid (Ni-NTA) agarose column (Ni-NTA, Qiagen, Hilden, Germany) equilibrated with the lysis buffer. The column was washed first with the wash buffer (Tris-HCl, pH 7.5, 300 mM NaCl, 25 mM imidazole) supplemented with 0.02% NP40 and second without NP40. Proteins were eluted with elution buffer (Tris-HCl, pH 7.5, 300 mM NaCl, 100 mM imidazole) [8,9,50]. Protein concentrations were determined by the Bio-Rad Protein Dye assay using BSA as a standard. The purity and concentrations were verified by electrophoresis on a 12% polyacrylamide Laemmli gel (Bio-Rad Mini-Protean; Hercules, CA, USA) containing sodium dodecyl sulfate (SDS-PAGE), and the proteins were visualized by staining the gels with Coomassie Brilliant Blue R-250. The purified proteins were stored in 50% glycerol at −80 °C until needed.

### 4.3. ATPase Assay

A colorimetric assay was used to measure the phosphate released during ATP hydrolysis based on Malachite green, as previously described in [8,9]. Reactions were performed at 37 °C in 96 well microtiter plates in a final volume of 50 μL. The reaction buffer contained 50 mM potassium acetate, 20 mM MES pH 6, 2 mM dithiothreitol (DTT), 0.1 mg/mL BSA and various concentrations of whole yeast RNA (Type XI-C, Sigma-Aldrich, St. Louis, MO, USA). 5 mM and 1 mM magnesium acetate were added to the LieIF4A and eIF4A_Mus_ reactions, respectively. Reactions were initiated by adding ATP; when not noted otherwise, the ATP concentration used was 1 mM. Reactions were stopped after various times with ethylenediaminetetraacetic acid (EDTA, final concentration 60 mM). The absorption at 630 nm was converted to phosphate concentration by a reference curve generated from a dilution series of a known phosphate concentration (0.1 mM Pi standard; Innova Biosciences, Cambridge, UK). Reaction velocities were determined by a linear regression fit of the phosphate generated against time. Only the initial linear phases of the curves were used. Velocities were determined for three independent experiments for each reaction condition. The Michaelis –Menten (K_m_) and the inhibition kinetics (K_i_) constants were obtained by nonlinear regression using GraphPad Prism 8 (GraphPad Software).

### 4.4. Parasite and Cell Cultures

The MON1 *L. infantum* laboratory strain LV50 originating from a visceral leishmaniasis case (Laboratory of Molecular Epidemiology and Experimental Pathology, Institut Pasteur de Tunis) was used to test the effect of the chemical compounds. Promastigote cultures were inoculated in RPMI-1640 media supplemented with 2 mM L-glutamine, 100 U/mL penicillin, 100 U/mL streptomycin and 10% (*v*/*v*) heat-inactivated fetal bovine serum at 22 °C. Promastigotes were collected at the beginning of the stationary phase (day 5), counted, centrifuged and seeded at 10^6^/mL in 10 mL of complete media [7].

The human myelomonocytic cell line, THP-1, was ordered from the American Type Culture Collection (ATCC, TIB-202). Cells were maintained in RPMI 1640/Glutamax-I media (Gibco BRL, Bleiswijk, The Netherlands) supplemented with 10% heat-inactivated fetal calf serum (Gibco, Bleiswijk, The Netherlands) plus penicillin G (100 U/mL) and streptomycin (100 g/mL). THP-1 cells were differentiated to macrophages after their treatment with 20 ng/mL phorbol 12-myristate 7-acetate (PMA) (Sigma, St. Louis, MO, USA) for 48 h at 37 °C, 5% CO2. The viability of THP-1 mature macrophage-like cells was determined to be >97% by the Trypan blue dye exclusion assay.

### 4.5. MTT Assays and Analysis for the Parasite Viability and Compound Toxicity on Macrophages

The effect of the chemical compounds on the viability of *L. infantum* promastigotes was evaluated by a colorimetric MTT (3-(4,5-dimethylthiazol-2yl)-2,5-diphenyl tetrazolium bromide) 96-well plate assay as previously described in [7]. This test consisted of a reduction of tetrazolium salt to a soluble crystal (blue formazan) by the succinate dehydrogenase activity of mitochondria in living cells, which can be quantified by spectrophotometry [51]. Briefly, 90 μL of LV50 promastigotes harvested from the stationary growth phase of 10^6^/mL culture were added to a 96-well culture plate with various concentrations of the selected compounds to obtain a final volume of 100 µL, compound concentrations of 1.5–100 μM, and a final concentration of 1% DMSO per well. Plates were incubated at 25 °C for 24 h.

For the cytotoxicity assay, we used the THP-1-derived macrophages (50,000 cells/well) seeded in 96-well plates and treated with serial concentrations of compounds at the same conditions as for the promastigotes for 24 h.

Mock-treated THP-1-derived macrophages and promastigotes (1% DMSO) were used as controls. After 24 h of incubation, MTT solution was added at 1 mg/mL to each well and incubated at the appropriate cell temperatures for 4 h. Then, 150 μL of DMSO was added to each well to dissolve the blue formazan, and the optical density (OD) was measured at 560 nm with a microplate reader (MULTISCAN, Labsystems, Thermo Fisher Scientific, Ratastie, Finland).

The promastigotes and THP-1-derived macrophage viabilities were expressed as the percentage of the viable promastigotes or cells in treated conditions relative to the 1% DMSO cultures, as previously described in [7]. The half maximal inhibitory concentration (IC_50_) and 50% cytotoxic concentrations (CC_50_) values were calculated for all the tested compounds by interpolation.

### 4.6. Statistical Analysis

The data presented in this study corresponded to the mean ± standard error (SE) of three independent experiments. Statistical analyses were performed using GraphPad Prism 8 statistical software program (GraphPad Software).

### 4.7. Molecular Docking

We used previous data on the most viable coordinates to define the potential binding site of 6-aminocholestanol and its analogs [8]. The receptor and ligand files were prepared using autodock tools [52]. For each molecule, hydrogen bonds were added, and gasteiger charges were calculated to obtain the pdbqt files required for the docking simulations. We generated the energy map files using autogrid4.

Docking was performed using autodock4 [52]. The genetic algorithm was used as a search algorithm along with the scoring function based on simulated annealing. A maximum number of 10 poses were generated for each compound. Pose clustering was performed. Most populated clusters were retained as the most pertinent, and the pose with the lowest score within these clusters was selected for each compound.

## 5. Conclusions

This study revealed cholesterol derivatives as a novel scaffold structure of eIF4A inhibitors that could have a higher affinity for *Leishmania* eIF4A than cholestanol derivatives previously described. Notably, 7-α-aminocholesterol exhibited higher efficiency against LieIF4A and *L. infantum* promastigotes viability and no toxicity. This class of compounds could be considered a promising route to screen and design more selective and efficient anti-*Leishmania* molecules targeting LieIF4A. Rocaglamide was found to be less effective on LieIF4A than on eIF4A_Mus_. The study confirmed that *Leishmania* and mammalian eIF4A proteins could be differently targeted and affected by effectors that inhibit or enhance their biochemical activities. These structural differences supported the prospect of identifying compounds selectively targeting *Leishmania* eIF4A without affecting the mammalian counterpart. Accordingly, this work further supported the chemical validation of LieIF4A as a favourable drug target against Leishmaniases.

## Figures and Tables

**Figure 1 molecules-27-05890-f001:**
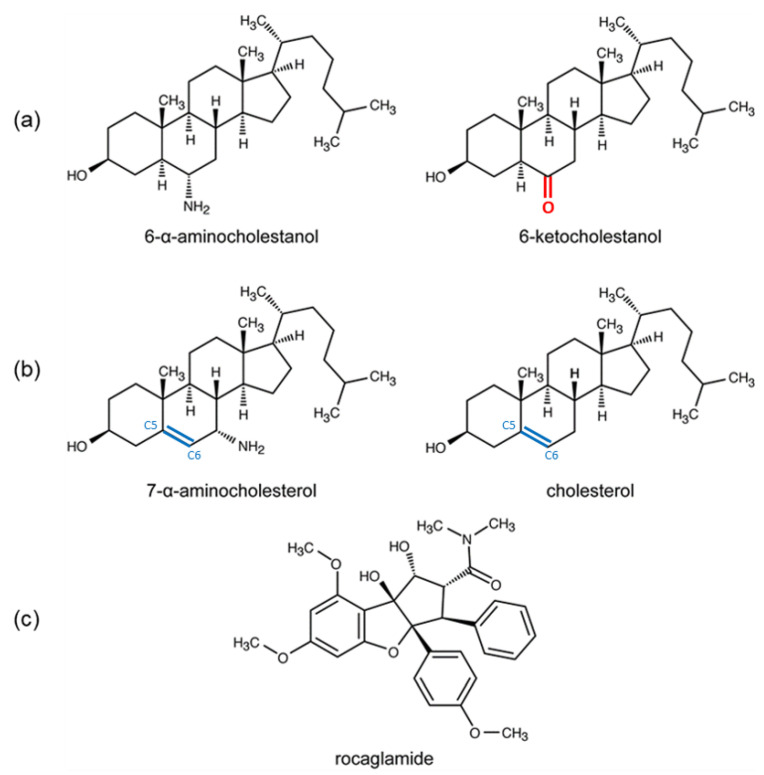
Chemical structures of the compounds used in this study. Chemical structures tested on the LieIF4A ATPase activity (**a**) cholestanol derivatives (6-α-aminocholestanol and 6-ketocholestanol), (**b**) cholesterol derivatives (7-α-aminocholesterol and cholesterol), (**c**) and rocaglate derivative (rocaglamide). The 6-ketocholestanol presents a keto group at position 6 (**in red**). The sterols present double bond between carbons 5 and 6 (**in blue**).

**Figure 2 molecules-27-05890-f002:**
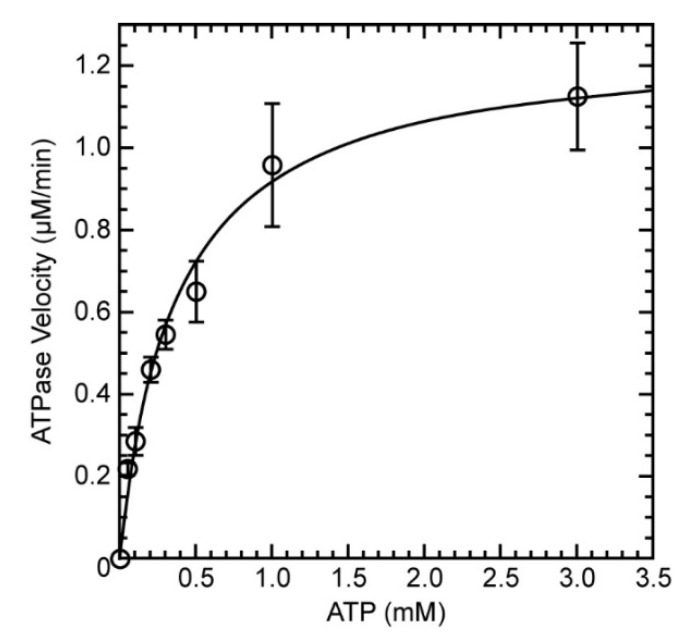
ATPase reaction velocities of eIF4A_Mus_ with 340 ng/μL yRNA and different concentrations of ATP. The means and standard deviations are shown for three independent experiments. The values were fit to the nonlinear Michaelis-Menten equation using GraphPad Prism software. All the reactions were performed in the presence of 10% DMSO and 820 nM LieIF4A.

**Figure 3 molecules-27-05890-f003:**
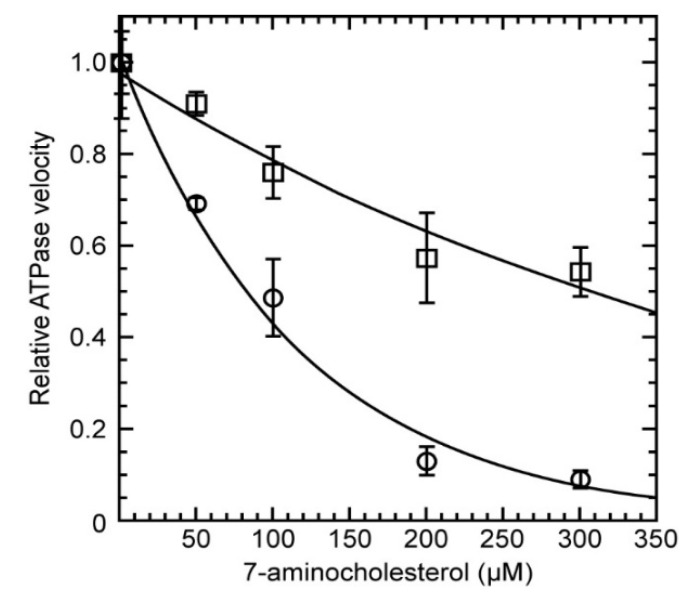
Relative ATPase reaction rates of the two proteins in the presence of increasing concentrations of 7-α-aminocholesterol and 340 ng/μL RNA. Mammalian eIF4AI was used at 1200 nM and LieIF4A at 820 nM. The error bars represent the mean and standard deviations of three independent measurements made. The relative reaction velocities were normalized to one in the absence of the inhibitor compound to facilitate comparisons. Data were fit to an exponential decay for visualization purposes and IC_50_ calculation. The relative reaction rate of LieIF4A (◌) and eIF4A_Mus_ (□) were plotted as a function of compound concentrations. The 7-α-aminocholesterol had different kinetic effects on the two proteins and clearly showed stronger inhibition on the *Leishmania* protein.

**Figure 4 molecules-27-05890-f004:**
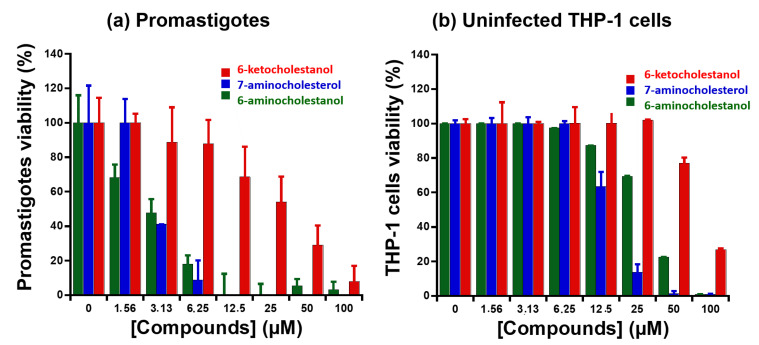
MTT cell viability assay shows a promising anti-leishmanial activity of the cholestanol and sterol derivatives in a dose-dependent manner and a little cellular toxicity at the active concentrations. (**a**) Effects on *L. infantum* promastigotes. (**b**) Effects on uninfected THP-1-derived macrophages. The cholesterol was not active at the tested concentrations.

**Figure 5 molecules-27-05890-f005:**
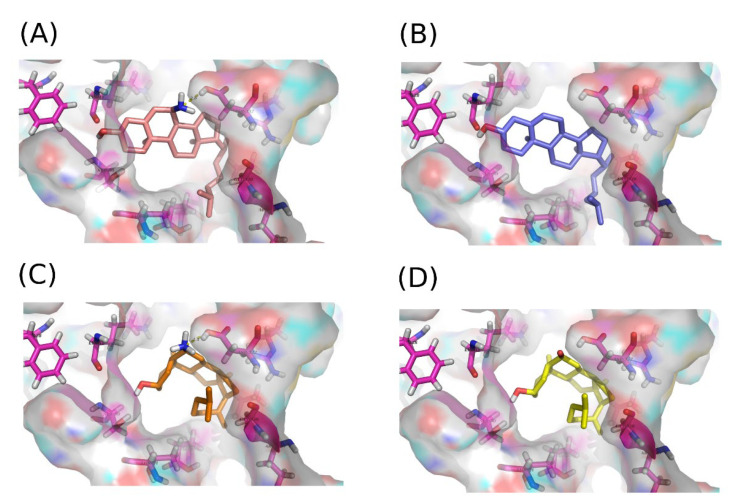
Docking poses of 7-α-aminocholesterol and its analogs on LieIF4A protein. Best scored docking pose of (**A**) 7-α-aminocholesterol, (**B**) cholesterol, (**C**) 6-α-aminocholestanol and (**D**) 6-ketocholestanol. Compounds 7-aminocholesterol and 6-α-aminocholestanol established a stable H-bond with Asp332 from motif V, shown in dashed lines with their estimated length in Å.

**Figure 6 molecules-27-05890-f006:**
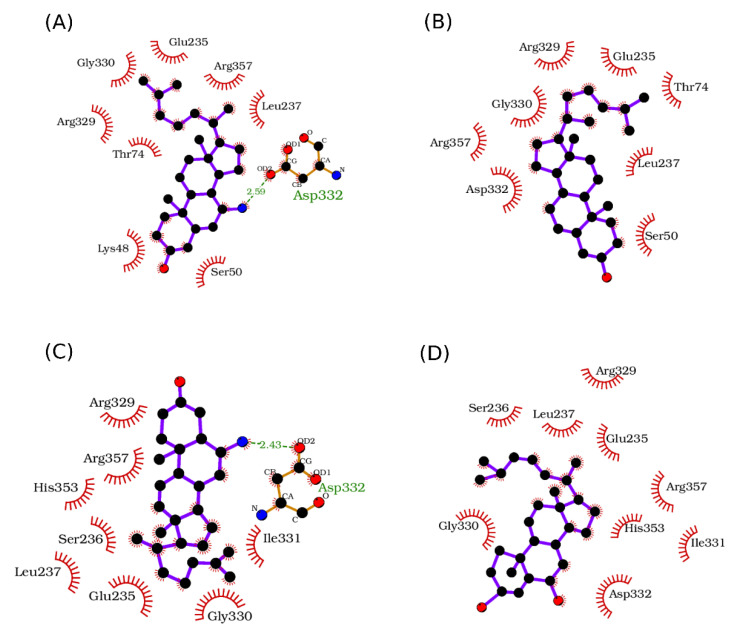
Protein-inhibitor interaction diagrams as determined with LigPlot+. Residues involved in hydrophobic interactions are shown as red half-circles, and H-bonds are shown in green dotted lines along with their lengths in Angstroms. (**A**) Diagram of interaction between LieIF4A and 7-α-aminocholesterol. (**B**) Diagram of interaction between LieIF4A and cholesterol. (**C**) Diagram of interaction between LieIF4A and 6-α-aminocholestanol. (**D**) Diagram of interaction between LieIF4A and 6-ketocholestanol.

**Table 1 molecules-27-05890-t001:** Rocaglamide has different effects on the ATPase activity of LieIF4A and eIF4A_Mus_.

Protein ^a^	[yRNA] (ng/µL) ^b^	Cmpd (µM) ^c^	V (µM/min) ^d^	Relative ^e^
eIF4A_Mus_	340	0	0.96 ± 0.15	1.00 ± 0.16
eIF4A_Mus_	55	0	0.41 ± 0.06	0.43 ± 0.06
eIF4A_Mus_	55	50	2.55 ± 0.03	2.65 ± 0.03
LieIF4A	340	0	1.76 ± 0.04	1.81 ± 0.05
LieIF4A	1	0	0.47 ± 0.07	0.49 ± 0.07
LieIF4A	1	50	0.77 ± 0.07	0.80 ± 0.07

^a^ Mouse eIF4A (P60843) was used at 1200 nM and LieIF4A (A4HRK0; LinJ_01_0800) at 820 nM. ^b^ Whole yeast RNA, type XI-C (Sigma-Aldrich, St. Louis, MO, USA), was used. The RNA concentrations were adjusted to give approximately the same ATPase activity for LieIF4A and eIF4A_Mus_ at suboptimal concentrations of RNA. Maximum reaction velocities were obtained with 340 ng/µL. Note that eIF4A_Mus_ has the same aminoacid sequence as human eIF4AI. ^c^ Rocaglamide A (Sigma-Aldrich, St. Louis, MO, USA) was dissolved in DMSO. ^d^ The mean velocities were determined from three or more independent experiments. The standard deviations around the mean are as shown. All reactions were done in the presence of 10% DMSO. ^e^ The activity relative to the maximum velocity of eIF4A_Mus_ in the absence of rocaglamide A. The standard deviations were calculated from the relative values.

**Table 2 molecules-27-05890-t002:** Summary of the K_i_ value calculations of the sterol and cholestanol derivatives for LieIF4A and eIF4A_Mus_.

Protein	K_m_ (µM)	Compound	K_i_ (µM)
LieIF4A	150 *	7-α-aminocholesterol	8.57 ± 1.61
		6-α-aminocholestanol	19.5 ± 3.27 *
		6-ketocholestanol	315.9 ± 80.65 *
		Cholesterol	NA
eIF4A_Mus_	351.78	7-α-aminocholesterol	80.48 ± 12.52
		6-α-aminocholestanol	20.22 ± 9.47 *
		6-ketocholestanol	484.5 ± 94.46 *
		Cholesterol	NA

The mean of Ki values was determined from three independent experiments. All experiments were done at the same time under the same conditions. The standard deviations around the mean were as shown. * Calculated from data in [7]. NA: not active.

**Table 3 molecules-27-05890-t003:** Summary of the in vitro effects of the cholestanol and sterol derivatives on the viability of the promastigotes and noninfected THP-1 cells.

Compounds	Promastigotes LV50	Uninfected THP-1-Derived Macrophages
	IC_50_ (µM)	CC_50_ (µM)
7-α-aminocholesterol	2.29 ± 0.4	17.5 ± 3.4
6-α-aminocholestanol	3.64 ± 0.1 *	35.2 ± 1.3 *
6-ketocholestanol	37.0 ± 6.6	76.9 ± 2.1
Cholesterol	NA	NA

The mean of IC_50_ and CC_50_ values were determined from three independent experiments. The standard deviations around the mean were as shown. * Measured using data in [7]. NA: not active.

**Table 4 molecules-27-05890-t004:** Docking scores of docked compounds.

	7-α-aminocholesterol	6-α-aminocholestanol	6-ketocholestanol	Cholesterol
Docking score (kcal/mol)	−8.58	−7.28	−6.85	−5.41
N° of poses/cluster	8	7	8	10

The number of poses per cluster is the number of docking poses clustered with the selected pose out of 10.

## Data Availability

Not applicable.

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
