# Peer review of "Enzymatic and Molecular Characterization of Anti-Leishmania Molecules That Differently Target Leishmania and Mammalian eIF4A Proteins, LieIF4A and eIF4AMus"

_molecules, 2022, doi:10.3390/molecules27185890_

Round 1

Reviewer 1 Report

In the manuscript “Enzymatic and molecular characterization of anti-Leishmania molecules that differently target Leishmania and Mammalian eIF4A proteins, LieIF4A and eIF4AMus”, Abdelkrim and co-workers identified 7-α-aminocholesterol as an inhibitor of Leishmania infantum eIF4A-like protein (LieIF4A), by interfering with the ATPase activity of LieIF4A. Importantly, 7-α-aminocholesterol has anti-leishmanial activity on L. infantum promastigotes whereas the affinity/inhibitory effect on eIF4A from mouse is about 10-fold lower (Ki: 8.57 µM ± 1.61 for LieIF4A and 80.48 µM ± 12.52 for eIF4AMus). Therefore, 7-α-aminocholesterol is a first hit in the drug development pipeline for a compound that specifically target eIF4A from Leishmania. Moreover, the authors analysed the anti-leishmanial activity for the rocaglate RocA, a plant-derived compound that interferes with the unwinding activity of eIF4A during translation initiation by clamping the 5´-UTR of selected mRNAs on the surface of eIF4A.

In general, the manuscript is well written and in principle, data are important, of relevance for the scientific community and suitable for publication in “Molecules”. However, there are more experimental data required to improve the study before publication is possible. In the current version the manuscript is rather incomplete and cannot be published. Several experiments that need to be done are listed below:

Major points

·        Include the human eIF4A1 in all experiments of this study. Leishmaniases is a neglected human tropical disease and a comparison with the human eIF4A1 enzyme is required

·        Section 2.1 - ATP hydrolysis: Compare the rate of phosphate release during ATP hydrolysis for eIF4AMus, LieIF4A and eIF4AHuman in the same experimental setup. Don´t refer to the previous study [8].

·        Table 1: What is the reason to use 1200 nM eIF4AMus and 820 nM LieIF4A? Use also higher and lower concentrations of the isolated eIF4A proteins (titration)

·        Section 2.2: For analysis of Roc A effects, it is more informative to use Thermal Shift Assay or Fluorescence Polarisation Assay instead of ATPase assay, since these assays reflect direct binding of Roc A to the eIF4A-RNA complex (RNA clamping)

·        Table 3 and section 2.4: When does the compounds 7-α-aminocholesterol, 6-α-aminocholestanol and 6-ketocholestanol start to get cytotoxic in uninfected cells? Provide the CC50 values in uninfected THP-1-derived macrophages

·        Fig. 4: Provide better docking poses (clear indication of Asp332 and the distance of H-bonds). Please explain why the double-bond between C5 and C6 in 7-α-aminocholesterol is important. This cannot be explained from the shown docking poses.

·        As mentioned in the discussion (line 446) the docking simulations focused on a zone of LieIF4A that include key residues of the conserved motifs of the RNA helicase [8].

Test the compounds (at least 7-α-aminocholesterol) also on their effect on helicase activity

Minor points

·        Introduction, line 100: the anti-leishmanial concentration of Roc A to inhibit L. infantum promastigotes was calculated at an EC50 of 16.5 µM. This seems to be highly unspecific since e.g. antiviral activity against several viruses for other rocaglates was calculated at low nanomolar concentrations (see Taroncher-Oldenburg et al., Microorganisms. 2021 Mar 5;9(3):540). At nanomolar concentrations rocaglates have potent cytotoxic effects in cancer cell lines (use as anti-tumor drugs). This should be clearly mentioned here.

·        Line 108/109: Rocaglates should inhibit ATP-dependent unwinding in wildtype eIF4A. Please check this.

·        Figure 3: Please include the concentrations of the two eIF4A enzymes used in the ATPase reaction

·        Figure 4 A+B): Is there a steric clash between the polar OH-group at C3 of 7-α-aminocholesterol or cholesterol and the negatively charged AA (what Amino acid is at this position?) possible?

·        Discussion, line 438: “the in vitro anti-leishmanial activity of 7-α-aminocholesterol against intramacrophage amastigotes of L. donovani (LV9) was studied, and it exhibited an IC50 value of 1.31 μM [39].”

Why is the effect on L. donovani better than on the separated protein?

Reviewer 2 Report

Review for:

“Enzymatic and molecular characterization of anti-Leishmania molecules that differently target Leishmania and Mammalian eIF4A proteins, LieIF4A and eIF4AMus”

Authors: Yosser Zina Abdelkrim Ep Guediche * , Emna Harigua-Souiai , Imen Bassoumi-Jamoussi , Mourad Barhoumi , Josette Banroques , Khadija Essafi-Benkhadir , Michael Nilges , Arnaud Blondel , N. Kyle Tanner , Ikram Guizani *

This manuscript is a follow-up to the 2018 paper, The steroid derivative 6-aminocholestanol inhibits the DEAD-box helicase eIF4A (LieIF4A) from the Trypanosomatid parasite Leishmania by perturbing the RNA and ATP binding sites by the same group. In the new 2022 manuscript, the authors compare kinetic parameters of eIF4AMus and LieIF4A and compare the inhibitory profiles of cholesterol and 7-?-aminocholesterol to the previously investigated 6-?-aminocholestanol. The group finds that 7-?-amino cholesterol, and not cholesterol, moderately inhibits eIF4AMus and LieIF4A ATPase activity, demonstrating the importance of the amino group to the inhibitory profile. Notably, 7-?-amino cholesterol inhibits LieIF4A ATPase activity with a ~10-fold higher potency than eIF4AMus, which is not the case for the previously investigated 6-?-aminocholestanol. 7-?-aminocholesterol also inhibits viability of L. infantum promastigotes, with a potency about 7-fold higher than THP-1-derived macrophages. The authors then dock the investigated compounds to LieIF4A and propose important interactions between the amino groups of 7-?-amino cholesterol and 6-?-aminocholestanol with the proposed LieIF4A binding pocket.

Although the authors have identified a molecule that is possibly worth additional investigation as a LieIF4A inhibitor, I do not think the manuscript provides significant progress from the previous manuscript or contributes to the field enough to warrant publication in its current state. Much of the data used in this paper was published in the aforementioned 2018 manuscript, and in addition, I believe there notable scientific flaws this publication that weaken the manuscript. I have listed some thoughts below:

1.     Introduction paragraph 2 (line 52) broadly describes various biological roles of eIF4A, but leaves out its main function, which is the enzymatic driving of cap-dependent translation. More specifically, eIF4A unwinds RNA secondary structure upstream in the 5’-untranslated region to provide space for the 43S pre-initiation complex to bind, scan for the start codon, and initiate cap-dependent translation. Addition of this information would provide a better framework for the function of eIF4A and give the reader better reference to why LieIF4A is a good drug target.

2.     The authors perform their enzymatic experiments in 10% DMSO, which is an extremely harsh condition for biological proteins.

3.     7-?-aminocholesterol inhibits viability of L. infantum promastigotes, but the IC50 is 4-fold higher than the Ki of LieIF4A ATPase inhibition, suggesting that the potency in L. infantum promastigotes could be due to an off-target effect. In addition, although the authors claim that 7-?-aminocholesterol shows no significant inhibition against THP-1 derived macrophage (line 289), it does inhibit viability of these macrophages with moderate potency. The IC50 of 7-?-amino cholesterol against THP-1 derived macrophages is only about 2-fold higher than the Ki of LieIF4A ATPase inhibition. This window is very tight, and at a dose needed to greatly inhibit viability of L. infantum promastigotes, this molecule would likely show off-target toxicity in human cells.

4.     Comparing LieIF4A to human eIF4A instead of eIF4AMus would strengthen the manuscript and provide more meaningful conclusions that these molecules may be able to inhibit LieIF4A while leaving human eIF4A unaffected.

5.     The authors provide no evidence to support docking results. It is unconfirmed if these molecules are being docked into the correct binding pocket or if Asp332 is important to LieIF4A inhibition by these molecules. Mutagenesis experiments with Asp332 would support the claim that the amino group on the inhibitors makes important interactions with this residue. This information could also help jumpstart a medicinal chemistry campaign to find more potent LieIF4A inhibitors.

6.     Data to support why 7-?-aminocholesterol was selected for investigation over other compounds would provide additional enthusiasm that 7-?-aminocholesterol is an exciting molecule that is more worthy of further investigation for LieIF4A inhibition than other cholesterol or cholestanol derivatives.

7.     A notable amount of the data, including ATPase IC50 values of 6-?-aminocholestanol and 6-ketocholestanol reported in this manuscript are from the 2018 paper. This indicates that these compounds were not tested side-by-side with 7-?-aminocholesterol, and the differences in Ki or IC50 may be due to different batches of protein, experimental conditions, or different researchers conducting the experiments. This also applies to kinetic parameters of LieIF4A which were used to determine yeast RNA concentrations to use in kinetic assays in this paper.

8.     The manuscript would benefit from dose response experiments being shown as dose response curves, instead of just reporting the IC50 values in a table.

9.     Figures are referred to out of order, notable carbons in Figure 1 should be labeled, and residues in docking in Figure 4 should be more clearly labeled.

I would recommend future investigation focus on why 7-?-aminocholesterol is able to inhibit LieIF4A ATPase activity with 10-fold greater potency than eIF4AMus, while 6-?-aminocholestanol inhibits both eIF4A proteins with similar potencies. I would be curious if this trend holds true with human eIF4A instead of mouse eIF4A.

Overall recommendation: Reject

Round 2

Reviewer 1 Report

The manuscript has been improved by the authors and is now in a suitable form  for publication

Reviewer 2 Report

The authors addressed many of the comments. For future assays and screening for new compounds, I would still suggest lowering the DMSO concentration. Although compounds may be relatively insoluble in water, 10% DMSO is a large amount that can affect protein activity. I would also suggest development of a direct-binding assay to test if compounds are able to bind eIF4A-WT and mutants (specifically a D332 mutant). This would strengthen the claim that the amino group on the inhibitors makes important interactions with this residue, and will be useful in further investigation.